# Adolescents’ Nutrition: The Role of Health Literacy, Family and Socio-Demographic Variables

**DOI:** 10.3390/ijerph192315719

**Published:** 2022-11-25

**Authors:** Stefano Delbosq, Veronica Velasco, Cecilia Vercesi, Luca Piero Vecchio

**Affiliations:** 1Psychology Department, Milano-Bicocca University, 20126 Milan, Italy; 2DG Welfare, Regione Lombardia, 20124 Milan, Italy

**Keywords:** health literacy, adolescent obesity, nutrition, food intake, family, socio-ecological model, social inequalities

## Abstract

Adolescent obesity rates are increasing on an epidemic level and food intake is one of the most important causes of this condition. From an ecological perspective, food intake is, in turn, influenced by many factors that need to be considered. This study aims to evaluate the associations between socio-demographic factors (gender, family origin, socio-economic status, parent’s education level), which consist of social stratifiers, health literacy and family context, as independent variables, and food intake (consumption of fruits, vegetables, soft drinks and sweets and breakfast frequency) and outcomes (Body Mass Index category), as dependent variables. Data were retrieved from 2145 students (13 and 15 years old) from the Lombardy region (Italy) who participated in the 2018 edition of Health Behaviour in School-Aged Children (HBSC). Six multiple binary logistic regression models were used in this study. Fruit, vegetable and soft drinks consumption models were related to all three-factor levels. Breakfast consumption frequency was associated with socio-demographic variables. BMI category was associated with socio-demographic and family variables. The results confirmed the existence of social inequalities, the importance of health literacy in predicting healthy behaviours and the relevance of the family context. The study confirms the importance of the ecological approach to understanding food intake and overweight/obesity status in adolescents.

## 1. Introduction

Global trends indicate the increase in obesity and overweight rates is an epidemic emergency for adults, children and adolescents [1,2,3]. Evidence suggests that the COVID-19 pandemic may have worsened the situation, reducing the possibility of physical activity and changing the behaviours and eating habits of children and adolescents [4,5]. Children and adolescents’ obesity is associated with several negative outcomes for both physical (e.g., increased risk of cardiovascular diseases or Type 2 Diabetes Mellitus) and mental (e.g., depression, anxiety, low self-esteem and eating disorders) health [2,6,7,8,9,10,11]. Crucially, obese children and adolescents tend to remain obese in adulthood, indicating that prevention measures should be considered of primary importance [10,12].

Obesity has multiple etiological factors, constituting the outcome of the interaction of biological, genetic, developmental, environmental and behavioural factors [9,10,11]. Like in the case of many other behaviours and outcomes, the factors leading to childhood and adolescent obesity can be included in the theoretical framework of the socio-ecological model (SEM) [13,14]. SEM includes the following levels of influence: individual (e.g., genetic and biological characteristics, knowledge, attitudes, beliefs and behaviours), interpersonal (e.g., family, peers, social networks and associations), institutional (e.g., rules, regulations and organisations), community (e.g., social networks, norms and standards) and policy (e.g., local, national and international policies and laws) level [14].

Food intake, together with physical activity, is one of the most important causes of children and adolescents’ overweight and obesity status: such conditions are related to an energy imbalance for a continuous time so that the caloric intake exceeds the energy consumed [9]. An Italian cohort study from 2010 to 2013 found that ultra-processed food consumption constituted 25.9% of average energy intake in children and adolescents, compared to an average of 17.3% among adults [15]. Ultra-processed food is defined as products made through physical, biological and chemical processes, typically with multiple ingredients and additives, including food such as soft drinks, sweets, processed meats and pre-prepared frozen meals [16,17]. Ultra-processed food consumption has been associated with obesity and adiposity parameters in longitudinal and cross-sectional studies [17,18]. Consumption of this kind of food has also been associated with other health risks, such as metabolic and cardiovascular diseases, depression and anxiety symptoms [18,19].

From an ecological perspective, food intake is, in turn, influenced by many factors that need to be considered in developing effective predictive and intervention models [14]. In particular, the literature indicates the likely influence of socio-demographic factors, individual skills and family context, as summarised in the following paragraphs. International systematic reviews of socio-demographic determinants of children and adolescent food intake are scarce. In the literature, some studies specifically linked socio-demographic determinants and food intake, while others linked them directly with health outcomes such as overweight and obesity conditions. Below are presented studies considering both approaches. Gender has been identified as a relevant predictor of childhood obesity: boys are more likely to be overweight or obese [20]. International gendered data for adolescents’ obesity are scarcer, but a Risk Factor Collaboration estimate [21] indicates a higher male prevalence. Italian male adolescents are more likely to be overweight or obese than females [22]. Moreover, there is evidence that socio-economic status and parents’ educational level are relevant factors in predicting children’s overweight and obesity status, although they can have different and even opposite effects based on context (e.g., high-income countries vs. middle-income countries) [20]. Studies indicated that adolescents from low socio-economic status families show a higher prevalence of obesity and soft drink consumption [23,24,25,26]. Adolescents and children with lower-educated mothers were exposed to more fast-food outlets, although this did not affect Body Mass Index (BMI), according to a study from The Netherlands [27]. Finally, adolescents with a migrant background may have further risks: for example, in an Italian study, students with both parents from a foreign country had a higher probability of not consuming breakfast daily [28]. These results showed that socio-demographic variables are important social stratifiers and inequalities factors for food intake and obesity.

In line with the individual level of SEM, personal knowledge, skills and competencies in nutrition and health are relevant in determining behaviours and habits. Health literacy is an important construct, as it is considered by international bodies and studies as an individual social determinant of health. It has been shown to influence healthy behaviours, health and social services access, health outcomes, health-related inequalities, the ability to manage long-term healthcare conditions, and social capital [29,30,31,32,33,34,35]. Furthermore, health literacy supports empowerment, participation, and autonomous development [33]. Worl Health Organization (WHO) Glossary of Health Promotion defines health literacy as “the cognitive and social skills which determine the motivation and ability of individuals to gain access to, understand and use information in ways which promote and maintain good health” [36]. Evidence from the literature suggests children and adolescents’ health literacy is a predictor of healthy food intake and overweight and obesity. Chrissini and Panagiotakos’ review [37] revealed a positive association between low health literacy levels in children, their ability to develop and follow obesity preventive behaviours, and the risk of being overweight or obese. Chari and colleagues [38] found an association between adolescents’ obesity and adolescents’ low health literacy levels. Other studies showed the influence of parents’ health literacy on children’s food intake [39,40,41]. More recently, since health literacy is a general construct, the constructs of food literacy and nutrition literacy emerged [42] and evidence showed an association between food/nutrition literacy and obesity [43,44] and fruit and vegetable consumption [45,46].

On another socio-ecological level, the family represents the first agent of socialisation, influencing children and adolescents’ personal and social development. Family is the first setting in which children’s food habits and behaviours are shaped since parents, and especially the mother, constitute nutritional gatekeepers [47,48,49]. This influence may be direct, through the food prepared at home, or indirect, through behaviours, attitudes and skills that are transmitted to children and adolescents [48]. Family meal frequency has been linked to having a higher probability for children and adolescents to belong in the normal weight range, have healthier dietary and eating patterns, and have less occurrence of disordered eating [48,50,51]. These effects seem to be determined by different reasons: less TV time during meals, parental modelling of healthy eating, higher food quality, positive atmosphere, children’s involvement in the meal preparation and longer meal duration [48]. More generally, unhealthy behaviours can be influenced by two other kinds of family factors [52]. On the one hand, many studies showed the relevance of affection, support and positive communication. On the other hand, parents should promote autonomy and responsibility. On this issue, the concept of monitoring is particularly relevant, which refers to a set of behaviours encompassing attention, control and supervision that parents apply to their children [53]. Both kinds of factors—support and monitoring—can influence food intake. Dimitratos and colleagues [54] underlined the importance of relational aspects for adolescents’ diet and obesity. Costarelli and colleagues [41] considered monitoring as an indicator of positive parental feeding practices. Moreover, they also showed that positive parental feeding practices are influenced by the levels of parents’ health literacy.

This study aims to evaluate the association of different predictors related to food intake (the consumption of fruits, vegetables, two ultra-processed food such as soft drinks and sweets and breakfast frequency) and BMI category. Gender, family origin, socio-economic status, parents’ education level (socio-demographic variables and social stratifiers), health literacy (individual level variable), family meal frequency, family support, mother–child communication and mother’s monitoring (interpersonal level variables) were used as independent variables. To our knowledge, there is a lack of studies considering the effects of multiple factors at different socio-ecological levels on different adolescents’ nutrition outcomes.

## 2. Materials and Methods

### 2.1. Data Collection

Data used in this study derive from the Health Behaviour in School-Aged Children (HBSC) survey, carried out in the Lombardy region in northern Italy in 2018 [55]. HBSC is a cross-national and comprehensive survey about health behaviours, conditions and social determinants in students who are 11, 13 and 15 years old. It is promoted by the World Health Organization, and approximately 50 countries are involved in it. In Italy, the HBSC survey involves a national sample and representative regional samples for each region.

### 2.2. Participants

A representative sample of 13- and 15-year-old Lombardian students was used in this study. Eleven-year-old students were excluded since their version of the questionnaire did not include the health literacy scale and some family variables. Students were selected via a random cluster sampling of schools, followed by a random sampling of classes within each school. Only the students who answered all the items on the health literacy scale and at least one nutrition question were considered for the analysis. The final sample included 2145 students: 52.7% of the participants were female, and the students were equally distributed across the two age groups considered (1071 and 1074, for 13- and 15-year-olds, respectively). Table 1 reports the characteristics of the sample.

### 2.3. Measures

Socio-demographic, family and nutrition variables were measured and used in this study. For every variable, indicators were created based on instructions from their original source validation and the international HBSC study protocol or based on classifications from previous studies [56,57].

#### 2.3.1. Socio-Demographic Variables and Social Stratifiers

Gender. Gender was recorded with a 2-option item: male (0) and female (1).

National family origin. Students were asked what countries their parents were born in. Possible responses were: Italy, Romania, Albania, Morocco, the People’s Republic of China, Tunisia (the most common migrants’ birthplace for Italy) or others. After choosing “Other”, students could write the specific country. The national origin of students’ families was categorised as Italian family (0) when both parents were Italian and foreign/mixed family (1) when at least one parent had a different nationality.

Perception of socio-economic status. The socio-economic condition was measured through the Family Affluence Scale (FAS) [58], which measures adolescents’ perceptions of their family living conditions. The scale is an indicator of family affluence and includes six items about material resources, such as the number of bathrooms at home or cars (e.g., “Does your family own a dishwasher?”). For the analysis, the indicator was recoded into low, middle, and high levels according to a threshold determined by the HBSC national and international networks [59].

Parents’ education level. Students were asked about the education level of both parents. The possible responses ranged from Elementary school to graduation, with “I don’t know” as a possible response. A variable was created with the following categories: both parents holding a middle school diploma (1), at least one of the two parents holding a high school diploma (2) and at least one of the two parents graduated (3).

#### 2.3.2. Individual Variables

Health literacy. The Health Literacy for School-Aged Children (HLSAC) scale was used to measure self-reported health literacy [60,61,62]. The Italian version was validated by Velasco and colleagues [63]. The instrument includes 10 items (e.g., “Having good information regarding health”) on a 4-point Likert scale (1 = “Not at all true”; 4 = “Absolutely true”). The score was calculated with the mean of the items. Cronbach’s alpha was 0.802.

#### 2.3.3. Family Variables

Family meal frequency. The frequency of meals consumed with the family was measured by asking students about the number of family meals during the week. The possible responses were: “Never” (1), “Less than once a week” (2), “About once a week” (3), “Most days” (4) and “Every day” (5).

Family support. The 4-item family subscale of the Multidimensional Scale of Perceived Social Support (MSPSS) [64] was administered to measure family support (e.g., “I can talk about my problems with my family”). Answers were provided on a 7-point Likert scale ranging from 1 (very strong disagreement) to 7 (very strong agreement). The score was calculated with the mean of the items. Cronbach’s alpha was 0.909.

Mother–child communication. The quality of communication between students and their mothers was measured by a single item asking students “How easy is it for you to talk to your mother about things that really bother you?”. Answers were provided on a 5-point Likert scale ranging from 1 (“Very easy”) to 5 (“I never see or do not have this person”). The item was reversed in order to evaluate the quality of the relationship positively.

Mother’s monitoring. Monitoring by the mother (i.e., mother’s awareness of their children’s activities in multiple domains) was measured with a 5-item scale (e.g., “Your mother knows what you do in your free time”). Answers were provided on a 4-point Likert scale ranging from 1 (“She knows it very well”) to 4 (“I never see or do not have this person”). Items were reversed and the score was calculated with the mean of the items. Cronbach’s alpha was 0.777.

#### 2.3.4. Nutrition and BMI Variables

Nutrition variables, which in our study represent the outcome variables, were categorised based on international recommendations [65].

Fruits and vegetables. Fruit and vegetable consumption was measured by asking students how many times they eat fruits/vegetables within the week. Answers were provided on a 7-point Likert with the following options: “Never” (1), “Less than once a week” (2), “Once a week” (3), “2/4 days a week” (4), “5/6 days a week”, “Once a day every day” (6), “More than once a day” (7). A dichotomous variable, respectively for fruit and vegetable intake was created with 0 when consumption was lower than once a day and 1 with daily consumption.

Sweets and drinks. The consumption of sweets and drinks was measured by asking how many times respondents ate sweets/drinks a week. Answers were provided on a 7-point Likert with the following options: “Never” (1), “Less than once a week” (2), “Once a week” (3), “2/4 days a week” (4), “5/6 days a week”, “Once a day every day” (6), “More than once a day” (7). A dichotomous variable, respectively for sweets and soft drinks intake, was created with 0 with consumption up to once a week and 1 with multiple consumptions during the week.

Breakfast frequency. The frequency of breakfast consumption was measured by asking how many times students usually have breakfast during school days. Answers were provided on a 6-point Likert scale: “I never have breakfast on school days” (1), “One day a week” (2), “Two days a week” (3), “Three days a week” (4), “Four days a week” (5), “Five days a week” (6). A dichotomous variable was created with 0 when breakfast did not occur every day and 1 with daily breakfast consumption.

BMI category. The body mass index (BMI) was calculated by using self-reported height and weight. The BMI is the ratio of weight to height squared [66]. Following Cole and colleagues’ [66] tables, a dichotomous variable was created with 1 when students were not overweight or obese and 2 when students were overweight or obese.

### 2.4. Statistical Analysis

Data analysis was conducted using the version 28.0.0.0 (190) of the software IBM Statistical Package for Social Science (SPSS) supplied from the Milano-Bicocca University. Six multiple binary logistic regression models were conducted using nutrition outcomes (specific foods, breakfast frequency and BMI category) as dependent variables.

Blocks of independent variables were inserted as follows: block 1 included socio-demographic and social stratifiers variables (i.e., participants’ gender, national family origin, socio-economic status and parents’ education), block 2 included health literacy and block 3 included family variables (i.e., family meals frequency, family support, mother’s monitoring and mother–child communication). Nagelkerke’s R^2^ was used as an effect size measure. Significant changes in Nagelkerke’s R^2^ after each block were measured with the Omnibus test of model coefficients offered by SPSS (“Model” for Model 1, “Block” for Model 2 and Model 3). Model 0 is the baseline model predicting the most common outcome as default: if the Omnibus test is not significant with regards to “Model”, the proposed model does not have any additional explanatory value.

## 3. Results

### 3.1. Description of the outcome variables

The participants’ distributions on the outcome variables have been reported in Table 2.

### 3.2. Fruit Consumption

The models effectively predicted the consumption of fruits (see Table 3). Compared to Model 0 and to the previous models, every model significantly increased the predictive power. Fruit consumption had several statistically significant associated factors.

All socio-demographic variables were significant predictors: daily consumption of fruits was positively associated with gender (females having higher probabilities of consumption), national family origin (families with at least one non-Italian parent having higher probabilities of consumption), family socio-economic status (the higher it was, the higher the probabilities of consumption) and parents’ education level (the higher it was, the higher the probabilities of consumption). The role of socio-demographic factors did not change in the 3 models.

Health literacy was a significant predictor: the higher it was, the higher the probability of daily consumption of fruits. Its role did not change with the addition of family variables. Although a significant predictor, health literacy had a relatively small exponential function value (Exp (B) = 1.026 in Model 3).

Among the family variables, family meal frequency was found as a significant predictor: the more students reported having meals with family, the higher the probability of daily consumption. The other variables concerning the family context did not show any relationship with daily fruit consumption.

### 3.3. Vegetable Consumption

The models effectively predicted the consumption of vegetables (see Table 4). Compared to Model 0 and to the previous models, every model significantly increased the predictive power. Vegetable consumption had several statistically significant associated factors.

All socio-demographic variables were significant predictors except for national family origin. Daily consumption of vegetables was positively associated with gender (females having higher probabilities of consumption), family socio-economic status (the higher it was, the higher the probabilities of consumption) and parents’ education level (the higher it was, the higher the probabilities of consumption). The role of socio-demographic factors did not change in the 3 models.

Health literacy was a significant predictor: the higher it was, the higher the probability of daily consumption of fruits. Its role did not change with the addition of family variables. Although a significant predictor, health literacy had a relatively small exponential function value (Exp (B) = 1.045 in Model 3).

Between the family variables, family meal frequency was found as a significant predictor: the more students reported having meals with family, the higher the probability of daily consumption. As for fruit intake, the other family variables were not significant predictors.

### 3.4. Sweets Consumption

The models were not able to predict the consumption of sweets (see Table 5). Although gender and mother’s monitoring resulted as statistically significant predictors, Model 3 had not a statistically significant higher predictive power compared to Model 0, meaning that a model guessing the most common occurrence (high consumption of sweets) had the same predictive effectiveness of a model considering socio-demographic factors, health literacy and family factors.

### 3.5. Soft Drink Consumption

The models effectively predicted the consumption of soft drinks (see Table 6). Compared to Model 0 and to the previous models, every model significantly increased the predictive power. Soft drink consumption had several statistically significant associated factors.

All socio-demographic variables were significant predictors except for the socio-economic status. High consumption of soft drinks was negatively associated with gender (females having lower probabilities of high consumption) and parents’ education level (the higher it was, the lower the probabilities of high consumption), and positively associated with national family origin (families with at least one non-Italian parent having higher probabilities of high consumption). The role of socio-demographic factors did not change in the 3 models.

Health literacy was a significant predictor: the higher it was, the lower the probability of high consumption. Its role did not change with the addition of family variables. Although a significant predictor, health literacy had a relatively small exponential function value (Exp (B) = 0.976 in Model 3).

Between the family variables, and differently than in the previous cases, it was the monitoring by the mother that was found as a significant predictor: the higher it was, the lower the probability of high consumption. The remaining variables concerning the family context did not show a significant effect.

### 3.6. Breakfast Consumption Frequency

The models effectively predicted breakfast frequency (see Table 7). Model 1 statistically predicted breakfast frequency more accurately than Model 0 and Model 3 statistically predicted more accurately than Model 2, although Model 2 did not constitute an improvement over Model 1. Breakfast frequency was statistically associated with 2 factors. Gender was negatively associated with breakfast frequency: females had lower probabilities of having breakfast daily. Socio-economic status was positively associated with breakfast frequency: the higher it was, the higher the probability of having breakfast daily.

Adding health literacy as a predictive variable did not improve Model 1. The addition of family variables did not result in more predictive factors, although the overall model had greater fitness than Model 2.

### 3.7. Students’ BMI Category

The models effectively predicted the students’ BMI category (see Table 8). Model 1 statistically predicted the BMI category more accurately than Model 0 and Model 3 statistically predicted more accurately than Model 2, although Model 2 did not constitute an improvement over Model 1. Students’ BMI category was statistically associated with 3 factors.

Among socio-demographic variables, gender and parents’ education level were negatively associated with the BMI category: female students and students with highly educated parents had lower probabilities of being overweight or obese. The role of socio-demographic factors did not change in the 3 models.

The addition of health literacy as a predictive variable did not improve Model 1. Between the family variables, only family support was found as a significant predictor: the more students reported being supported by their families, the lower probability they had of being overweight or obese.

## 4. Discussion

This study used a representative regional sample to identify and measure the associations of different factors with food intake and BMI category of 13- and 15-year-old students. The results confirm the importance of adopting a Socio-Ecological approach with regard to food intake and overweight and obesity status in adolescents. All explanation levels used in this study proved their association with outcome measures, although to different degrees.

The results confirmed the importance of social stratifiers in determining different healthy eating behaviours. Gender was the factor with the most associations: females were more likely to report daily fruit and vegetable consumption and everyday breakfast consumption and less likely to report high soft drinks consumption. They were also less likely to be overweight or obese than males, which is consistent with the literature and the Italian adolescent population [22]. National family origin was a significantly associated factor of fruit and soft drinks consumption: students from families where at least one parent was not Italian were more likely to report high soft drinks consumption and, interestingly, daily fruit consumption. Nardone and colleagues [28] found that Italian adolescents with both parents from a foreign country had a higher risk of not consuming breakfast daily. In this study, no difference in breakfast consumption with regard to family origin emerged. This may be due to the different categorizations of migrant families: in our study, we compared families with both parents born in Italy with families with at least one parent from other countries while Nardone and colleagues [28] defined migrant families when both parents were from a foreign country. Moreover, the study of Nardone and colleagues [28] refers to a national sample including regions with different migrant histories. The associations between nutrition and national family origin should be better investigated. Socio-economic status was successfully associated with fruit and vegetable consumption, as well as breakfast consumption frequency: students from more affluent families were more likely to report healthy food intake. These results seem to reflect the social inequalities’ patterns found in the literature, although no relation was found with ultra-processed foods [20,23,24,25,26]. Parents’ education level was positively associated with fruit and vegetable consumption and negatively associated with soft drink consumption, indicating a social inequality for students with lower-educated parents. Students who reported having lower-educated parents were also more likely to be overweight or obese. These results partially confirmed data from the literature [20,27]. It is also relevant that the socio-demographic variables associations are significant in the models that include health literacy and family variables as well, showing the high relevance of social inequalities. This role of socio-demographic factors is also coherent with the results of Nardone and colleagues’ [28] study on the Italian adolescent population. Since social inequalities were proven to be central in determining healthy food intake, future studies should consider the effects of these factors in families with different socio-demographic backgrounds.

On an individual level, health literacy was found to be a significant factor associated with fruit, vegetable and soft drinks consumption: the higher health literacy levels, the more likely students were to report healthy food intake. These results are coherent with the literature highlighting the importance of health literacy as a predictor of healthy habits and behaviours [67]. Adding health literacy to those models always led to a significant increase in their explanatory capacity. It is worth noting, however, the small value of the B of the exponential function: although a significant factor, health literacy’s association was not high. These results seem to confirm the literature suggesting the use of subdimensions of health literacy when considering eating behaviours, such as nutrition literacy and food literacy [42]. Since health literacy is a general construct, these more specific constructs could be more appropriate in predicting healthy food intake and obesity. In fact, some studies have already related nutrition literacy and food literacy to these outcomes [43,44,45,46]. More research, comparing the predictive value of health literacy and food/nutrition literacy, is certainly needed. Furthermore, health literacy associations are significant in the models that include both health literacy and family variables, showing the relevance of this individual level.

The results at the family level show different processes through which the family may influence kids’ food intake. Family meal frequency was significantly associated with fruit and vegetable consumption, in line with the literature [48,50,51]. Probably, the importance of this kind of food is highly recognised and the common experience constitutes an opportunity to transmit food literacy, attitudes, behaviours and models. These results are also consistent with the association between health literacy and fruit and vegetable consumption. On the contrary, the consumption of soft drinks is associated with the parent’s ability to promote kids’ autonomy and responsibility through monitoring actions [53]. These results are also consistent with the association between health literacy and soft drink consumption. Finally, family support was a protective factor for overweight and obesity status, in line with the literature [68]. These results confirm the importance of considering both nutrition behaviours and emotional factors that influence overweight and obesity status. Actually, parents’ educational styles could pertain to the emotional and communication relationship between parents and adolescents, the autonomy-promoting role of the mother’s monitoring or the experience-sharing role of family meal frequency. These three processes through which the family may influence kids’ nutrition behaviours should be considered in designing nutrition parents’ training. Moreover, given the literature showing an association between parents’ health literacy and children’s healthy food intake [39,40,41], it would be relevant to investigate how parents’ health literacy influences both parents’ educational styles with regard to nutrition and children’s and adolescents’ health literacy.

The model predicting sweets consumption failed to demonstrate a statistically significant improvement over the default model. This result may be due to the disproportionated distribution of this variable (77.9% consumed sweets more than once a week) but the absence of associations should be investigated more. Another interesting result regards breakfast consumption. This behaviour was associated with socio-demographic variables (gender and socio-economic status), but not with health literacy and family variables. It seems that only social inequalities variables may influence daily breakfast consumption. With regards to socio-economic status, the relation could be explained by the fact that lower affluent families have fewer opportunities and time to consume breakfast together. Time restraints due to parental work demands (e.g., the need to leave home early in the morning) would limit the possibility of consuming breakfast together and transmitting important values, attitudes and behaviours. Several strengths and limitations of this study should be acknowledged. Since HBSC is a cross-sectional survey, the causal relationship between the factors and the outcomes has not been demonstrated. Longitudinal studies should be developed. Still, the models allowed to measure the associations of different factors on the outcomes and their conjunct effects on other variables. Second, although effect size values were relatively small, the addition of individual and family levels resulted in statistical improvement in the fitness of the models. Third, the study included data collected in just one Italian region. Other studies should consider data from wider geographical areas. However, the use of a regional representative sample attests to the validity of the study, and the Lombardy region is a very populated area with a number of inhabitants compared to many European countries. Lastly, although subdimensions of health literacy could constitute better predictive factors leading to healthy behaviours, the Health Literacy for School-Aged Children (HLSAC) scale used in this study has a solid background [60,61,62,63,69,70].

## 5. Conclusions

This study highlighted the importance of different factors associated with food intake related to adolescents’ overweight and obesity. In line with a socio-ecological model approach, associations between socio-demographic variables, individual level competence, such as health literacy, and interpersonal level family variables and fruit, vegetable and soft drinks consumption, as well as breakfast consumption frequency and BMI category were verified in a representative regional sample of adolescents from Italy. All these factors should be considered in order to treat and prevent adolescents’ unhealthy eating habits and the increasing overweight and obesity rates.

## Figures and Tables

**Table 1 ijerph-19-15719-t001:** Frequencies of socio-demographic and social stratifier variables.

Variable	Frequency (%)
	(*N* = 2145)
Age	
13 years old	1071 (49.9)
15 years old	1074 (50.1)
Gender	
Male	1015 (47.3)
Female	1130 (52.7)
Family Nationality	
Both parents from Italy	1759 (82.0)
At least one parent from another country	316 (14.7)
Missing answer	70 (3.3)
Family Affluence Scale (FAS)	
Low level	457 (21.3)
Middle level	1027 (47.9)
High level	617 (28.8)
Missing answer	44 (2.0)
Parents’ Education	
Up to middle school	452 (21.1)
At least one to high school	838 (39.1)
At least one with a degree	697 (32.5)
Missing answer	158 (7.3)

**Table 2 ijerph-19-15719-t002:** Frequencies of outcome variables.

Variable	Frequency (%)
	(*N* = 2145)
Fruit consumption	
Consumption lower than once a day (0)	1343 (62.6)
Daily consumption (1)	799 (37.2)
Missing answer	3 (0.1)
Vegetable consumption	
Consumption lower than once a day (0)	1377 (64.2)
Daily consumption (1)	763 (35.6)
Missing answer	5 (0.2)
Sweets consumption	
Consumption up to once a week (0)	471 (22.0)
Weekly consumption (1)	1671 (77.9)
Missing answer	3 (0.1)
Drinks consumption	
Consumption up to once a week (0)	1268 (59.1)
Weekly consumption (1)	875 (40.8)
Missing answer	2 (0.1)
Breakfast frequency	
Never or not always (0)	854 (39.8)
Every day (1)	1264 (58.9)
Missing answer	27 (1.3)
BMI category	
Not overweight/obese (1)	1555 (72.5)
Overweight/obese (2)	223 (10.4)
Missing answer	367 (17.1)

**Table 3 ijerph-19-15719-t003:** Results of multiple binary logistic regression models predicting fruit consumption.

Levels	Variables	Model 1		Model 2		Model 3	
		B (SE)	Exp (B)	B (SE)	Exp (B)	B (SE)	Exp (B)
Socio-demographic factors	Gender	0.452 (0.098) ***	1.571	0.451 (0.098) ***	1.570	0.455 (0.099) ***	1.577
	Family origin	0.299 (0.140) *	1.349	0.285 (0.140) *	1.330	0.325 (0.142) *	1.384
	FAS	0.243 (0.075) **	1.275	0.240 (0.075) **	1.271	0.237 (0.075) *	1.267
	Parent’s education	0.295 (0.070) ***	1.334	0.291 (0.070) ***	1.336	0.297 (0.071) ***	1.345
Individual factor	Health literacy			0.025 (0.011) *	1.025	0.026 (0.012) *	1.026
Family factors	Family meals frequency					0.273 (0.075) ***	1.314
	Family support					−0.043 (0.043)	0.958
	Mother–child communication					0.019 (0.062)	1.020
	Mother’s monitoring					0.008 (0.119)	1.008
	Nagelkerke’s R^2^	0.047		0.050		0.060	
	Omnibus test of model coefficients (df)	65.121 (4) ***		4.929 (1) *		14.567 (4) **	

Categorical variables: Gender (Male = 0, Female = 1), Family origin (Both parents from Italy = 0, At least one parent from other countries = 1). Ordinal variables: FAS (Low level = 1, Middle level = 2, High level = 3), Parent’s education (Up to middle school = 1, At least one to high school = 2, At least one with a degree = 3). Other variables are continuous. * *p* < 0.05; ** *p* < 0.01; *** *p* < 0.001.

**Table 4 ijerph-19-15719-t004:** Results of multiple binary logistic regression models predicting vegetable consumption.

Levels	Variables	Model 1		Model 2		Model 3	
		B (SE)	Exp (B)	B (SE)	Exp (B)	B (SE)	Exp (B)
Socio-demographic factors	Gender	0.751 (0.100) ***	2.119	0.754 (0.101) ***	2.162	0.755 (0.102) ***	2.127
	Family origin	0.089 (0.144)	1.093	0.061 (0.145)	1.063	0.096 (0.147)	1.101
	FAS	0.250 (0.076) **	1.284	0.245 (0.077) **	1.278	0.240 (0.077) **	1.272
	Parent’s education	0.328 (0.071) ***	1.389	0.323 (0.072) ***	1.381	0.329 (0.072) ***	1.390
Individual factor	Health literacy			0.047 (0.012) ***	1.048	0.244 (0.012) ***	1.045
Family factors	Family meals frequency					0.199 (0.075) **	1.220
	Family support					−0.013 (0.045)	0.987
	Mother–child communication					0.109 (0.063)	1.116
	Mother’s monitoring					0.089 (0.124)	1.093
	Nagelkerke’s R^2^	0.072		0.084		0.094	
	Omnibus test of model coefficients (df)	102.030 (4) ***		16.858 (1) ***		14.190 (4) ***	

Categorical variables: Gender (Male = 0, Female = 1), Family origin (Both parents from Italy = 0, At least one parent from other countries = 1). Ordinal variables: FAS (Low level = 1, Middle level = 2, High level = 3), Parent’s education (Up to middle school = 1, At least one to high school = 2, At least one with a degree = 3). Other variables are continuous. ** *p* < 0.01; *** *p* < 0.001.

**Table 5 ijerph-19-15719-t005:** Results of multiple binary logistic regression models predicting sweets consumption.

Levels	Variables	Model 1		Model 2		Model 3	
		B (SE)	Exp (B)	B (SE)	Exp (B)	B (SE)	Exp (B)
Socio-demographic factors	Gender	−0.195 (0.114)	0.823	−0.194 (0.115)	0.824	−0.229 (0.116) *	0.795
	Family origin	−0.063 (0.163)	0.939	−0.060 (0.163)	0.942	−0.050 (0.164)	0.951
	FAS	0.059 (0.087)	1.061	0.060 (0.087)	1.062	0.067 (0.088)	1.070
	Parent’s education	0.095 (0.221)	1.100	0.097 (0.081)	1.101	0.100 (0.081)	1.105
Individual factor	Health literacy			−0.006 (0.013)	0.994	−0.011 (0.013)	0.990
Family factors	Family meals frequency					−0.034 (0.083)	0.967
	Family support					−0.018 (0.050)	0.982
	Mother–child communication					−0.057 (0.071)	0.944
	Mother’s monitoring					0.320 (0.129) *	1.378
	Nagelkerke’s R^2^	0.005		0.005		0.011	
	Omnibus test of model coefficients (df)	6.219 (4)		0.216 (1)		6.981 (4)	

Categorical variables: Gender (Male = 0, Female = 1), Family origin (Both parents from Italy = 0, At least one parent from other countries = 1). Ordinal variables: FAS (Low level = 1, Middle level = 2, High level = 3), Parent’s education (Up to middle school = 1, At least one to high school = 2, At least one with a degree = 3). Other variables are continuous. * *p* < 0.05.

**Table 6 ijerph-19-15719-t006:** Results of multiple binary logistic regression models predicting soft drinks consumption.

Levels	Variables	Model 1		Model 2		Model 3	
		B (SE)	Exp (B)	B (SE)	Exp (B)	B (SE)	Exp (B)
Socio-demographic factors	Gender	−0.881 (0.097) ***	0.414	−0.880 (0.097) ***	0.415	−0.852 (0.098) ***	0.427
	Family origin	0.366 (0.139) **	1.442	0.384 (0.140) **	1.469	0.355 (0.142) *	1.426
	FAS	−0.014 (0.075)	0.986	−0.009 (0.075)	0.991	−0.012 (0.075)	0.988
	Parent’s education	−0.261 (0.070) ***	0.770	−0.256 (0.070) ***	0.774	−0.263 (0.070) ***	0.769
Individual factor	Health literacy			−0.029 (0.011) **	0.971	−0.024 (0.011) *	0.976
Family factors	Family meals frequency					−0.104 (0.070)	0.901
	Family support					0.032 (0.043)	1.033
	Mother–child communication					−0.004 (0.061)	1.005
	Mother’s monitoring					−0.370 (0.117) **	0.691
	Nagelkerke’s R^2^	0.074		0.078		0.087	
	Omnibus test of model coefficients (df)	105.354 (4) ***		6.920 (1) **		13.056 (4) *	

Categorical variables: Gender (Male = 0, Female = 1), Family origin (Both parents from Italy = 0, At least one parent from other countries = 1). Ordinal variables: FAS (Low level = 1, Middle level = 2, High level = 3), Parent’s education (Up to middle school = 1, At least one to high school = 2, At least one with a degree = 3). Other variables are continuous. * *p* < 0.05; ** *p* < 0.01; *** *p* < 0.001.

**Table 7 ijerph-19-15719-t007:** Results of multiple binary logistic regression models predicting breakfast consumption.

Levels	Variables	Model 1		Model 2		Model 3	
		B (SE)	Exp (B)	B (SE)	Exp (B)	B (SE)	Exp (B)
Socio-demographic factors	Gender	−0.444 (0.096) ***	0.641	−0.447 (0.096) ***	0.640	−0.463 (0.098) ***	0.629
	Family origin	−0.065 (0.139)	0.937	−0.077 (0.139)	0.926	−0.027 (0.141)	0.973
	FAS	0.162 (0.073) *	1.175	0.159 (0.073) *	1.172	0.150 (0.074) *	1.162
	Parent’s education	0.005 (0.068)	1.005	0.001 (0.069)	1.001	0.003 (0.069)	1.003
Individual factor	Health literacy			0.018 (0.011)	0.1018	0.010 (0.011)	1.010
Family factors	Family meals frequency					0.095 (0.069)	1.100
	Family support					0.074 (0.042)	1.077
	Mother–child communication					0.109 (0.071)	1.115
	Mother’s monitoring					0.194 (0.115)	1.214
	Nagelkerke’s R^2^	0.021		0.023		0.039	
	Omnibus test of model coefficients (df)	28.591 (4) ***		2.769 (1)		23.114 (4) ***	

Categorical variables: Gender (Male = 0, Female = 1), Family origin (Both parents from Italy = 0, At least one parent from other countries = 1). Ordinal variables: FAS (Low level = 1, Middle level = 2, High level = 3), Parent’s education (Up to middle school = 1, At least one to high school = 2, At least one with a degree = 3). Other variables are continuous. * *p* < 0.05; *** *p* < 0.001.

**Table 8 ijerph-19-15719-t008:** Results of multiple binary logistic regression models predicting BMI category.

Levels	Variables	Model 1		Model 2		Model 3	
		B (SE)	Exp (B)	B (SE)	Exp (B)	B (SE)	Exp (B)
Socio-demographic factors	Gender	−0.792 (0.160) ***	0.453	−0.785 (0.160) ***	0.456	−0.830 (0.162) ***	0.436
	Family origin	0.307 (0.215)	1.359	0.325 (0.216)	1.384	0.302 (0.219)	1.353
	FAS	−0.228 (0.119)	0.796	−0.217 (0.119)	0.805	−0.187 (0.121)	0.829
	Parent’s education	−0.240 (0.113) *	0.787	−0.236 (0.113) *	0.790	−0.257 (0.114) *	0.773
Individual factor	Health literacy			0.034 (0.018)	0.967	−0.029 (0.018)	0.971
Family factors	Family meals frequency					0.079 (0.113)	1.082
	Family support					−0.224 (0.064) ***	0.799
	Mother–child communication					0.009 (0.097)	1.009
	Mother’s monitoring					0.302 (0.199)	1.352
	Nagelkerke’s R^2^	0.047		0.051		0.071	
	Omnibus test of model coefficients (df)	39.435 (4) ***		3.645 (1)		16.572 (4) **	

Categorical variables: Gender (Male = 0, Female = 1), Family origin (Both parents from Italy = 0, At least one parent from other countries = 1). Ordinal variables: FAS (Low level = 1, Middle level = 2, High level = 3), Parent’s education (Up to middle school = 1, At least one to high school = 2, At least one with a degree = 3). Other variables are continuous. * *p* < 0.05; ** *p* < 0.01; *** *p* < 0.001.

## Data Availability

The data underlying this article were provided by the Lombardy region by permission. Data will be shared on request to the corresponding author with the permission of the Lombardy region.

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
