# Peer review of "Adolescents’ Nutrition: The Role of Health Literacy, Family and Socio-Demographic Variables"

_ijerph, 2022, doi:10.3390/ijerph192315719_

Round 1

Reviewer 1 Report

Comments to the editor

Thank you for the invitation to review “Adolescents’ Nutrition Behaviors: The role of health literacy, family and socio-demographic variables,” submitted to IJERPH. This is an important and timely research article. Just a few comments to the author.

Table 1: spell out FAS or have a footnote. Some of the percents do not add up to 100.

Why were 13 And 15 year olds selected; might want to justify that decision.

May want to insert a footnote on Table 3 indicating what the blocks indicate.

Since the discussion goes into detail, the conclusions could be shortened

Author Response

We thank you for your thoughtful suggestions and insights, which have helped us improved the paper significantly. We are also grateful for your positive feed-backs.

The manuscript has been rechecked and the necessary changes have been made in accordance with your suggestions. The responses to all comments have been prepared and given below.

Reviewer's comments:

Thank you for the invitation to review “Adolescents’ Nutrition Behaviors: The role of health literacy, family and socio-demographic variables,” submitted to IJERPH. This is an important and timely research article. Just a few comments to the author.

  • Table 1: spell out FAS or have a footnote. Some of the percents do not add up to 100.
  • Why were 13 And 15 year olds selected; might want to justify that decision.
  • May want to insert a footnote on Table 3 indicating what the blocks indicate.
  • Since the discussion goes into detail, the conclusions could be shortened

Response:

Thanks for the very useful suggestions, which have helped us improve the paper significantly. We changed the paper according to your suggestions:

  • FAS has been spelt out in Table 1. We added non-respondents percentages to Tables 1 and 2 so that they add up to 100.
  • We added a justification for the exclusion of 11 years old students from analysis, specifying that their version of the questionnaire didn’t include the health literacy scale and some of the family variables.
  • We added a footnote on the Tables reporting the results in order to make them easier to interpret and added a column specifying what the blocks indicate.
  • We shortened the conclusions following your suggestion.

Reviewer 2 Report

TO AUTHORS

The study is interesting, as it adds evidence in the literature on the association of individual characteristics of adolescents with their food consumption, eating habits and BMI. The text is well written. My contributions are focused on clarifying some parts of the text - especially with regard to the term "nutrition behaviours", which seems inappropriate to be used in the study.

1. ABSTRACT

The abstract is difficult to follow. I understand there is a word limit, but it is helpful to review the abstract. For example, let us know what the independent and dependent variables are to make the objective clearer. The reader does not yet know what the sociodemographic variables are, but there is something about social inequality; Would this be related to the income variable?

2. INTRODUCTION

2.1. I am not familiar with the term “nutrition behaviours” used in the title of the article and also throughout the text to refer to the consumption of fruits, vegetables, soft drinks and sweets and breakfast frequency. In the third paragraph of the Introduction, the authors reported for the first time in the text “nutrition behaviours” and used reference 9 for the background; however, this reference does not use “nutrition behaviours”. In this way, I suggest that the authors review the entire manuscript to change “nutrition behaviours” to “food intake”, as this seems to be more appropriate. Importantly, BMI is not a “nutrition behaviour”, but a result of food consumption.

2.2. Provide references [xx] that fit the following sentences: (i) “In particular, the literature [xx] indicates the likely influence of socio-demographic factors, individual skills and family context.” (ii) “In the literature, some studies [xx] specifically linked socio-demographic determinants and nutrition behaviours, while others [xx] linked them directly with health outcomes such as overweight and obesity conditions”.

2.3. Please review: “Health literacy is a construct considered an important individual social determinant of health by many international bodies and studies” for “Health literacy is an important construct, as it is considered by international bodies and studies as an individual social determinant of health”.

2.4. To make the objective clearer, inform what the independent and dependent variables were.

3. MATERIALS AND METHODS

Table 1 - what does FAS mean?

4. RESULTS

It is difficult to interpret Tables 3 to 8 without having text. Therefore, I suggest that the authors add a note to each Table to say which groups were compared for each variable.

5. DISCUSSION

5.1. I suggest deleting the following part of the Discussion: “Children and adolescents’ overweight and obesity rates (…) to shape and influence children and adolescents’ behaviours” (first paragraph); as it was already presented in the Introduction section. Once that's done, the first paragraph of the Discussion section might look like this: “This study used a regionally representative Italian sample of 13- and 15-year-old students to measure and identify associations of participants’ different characteristics with their food consumption and BMI. The results confirm the importance of adopting socio-ecological approaches with regards to food consumption and eating habit of students, as well as their weight status. All explanation levels used in this study proved their association with outcome measures, although to different degrees".

5.2. Nardone and colleagues’ [28] findings do not support the result that students from families where at least one parent was not Italian were more likely to consume fruit daily. What can the authors say about this to include in the discussion (e.g., a speculation)?

5.3. Authors’ sentence: “This behaviour was associated with socio-demographic variables but not with health literacy and family variables". Add a comma (,) before the word “but”.

5.4. Authors’ sentence: “It seems that only social inequalities variables may influence daily breakfast”. Is there a why?

6. CONCLUSIONS

Authors’ sentence: “This study highlighted the importance of different factors associated with nutrition behaviours related to children and adolescents’ overweight and obesity”. Did the study involve people of childhood age?

Author Response

We thank you for your thoughtful suggestions and insights, which have helped us improved the paper significantly. We are also grateful for your positive feed-backs.

The manuscript has been rechecked and the necessary changes have been made in accordance with your suggestions. The responses to all comments have been prepared and given below.

The study is interesting, as it adds evidence in the literature on the association of individual characteristics of adolescents with their food consumption, eating habits and BMI. The text is well written. My contributions are focused on clarifying some parts of the text - especially with regard to the term "nutrition behaviours", which seems inappropriate to be used in the study.

Response: We thank you for your positive feed-backs.

  1. ABSTRACT

The abstract is difficult to follow. I understand there is a word limit, but it is helpful to review the abstract. For example, let us know what the independent and dependent variables are to make the objective clearer. The reader does not yet know what the sociodemographic variables are, but there is something about social inequality; Would this be related to the income variable?

Response: Thanks for the very useful suggestions, which have helped us improve the abstract significantly. We revised the abstract to make it clearer. We listed the socio-demographic variables and we specified that these variables consist of social stratifiers. We also specified the independent and dependent variables.

  1. INTRODUCTION

2.1. I am not familiar with the term “nutrition behaviours” used in the title of the article and also throughout the text to refer to the consumption of fruits, vegetables, soft drinks and sweets and breakfast frequency. In the third paragraph of the Introduction, the authors reported for the first time in the text “nutrition behaviours” and used reference 9 for the background; however, this reference does not use “nutrition behaviours”. In this way, I suggest that the authors review the entire manuscript to change “nutrition behaviours” to “food intake”, as this seems to be more appropriate. Importantly, BMI is not a “nutrition behaviour”, but a result of food consumption.

Response: Thanks for the very useful suggestion, which has helped us improve the paper significantly. We replaced “nutrition behaviours” with “food intake” throughout the text. We used the term “nutrition” in the paper title to maintain a link with both food intake and BMI. As you wrote, BMI is not a behaviour but a result of nutrition.

2.2. Provide references [xx] that fit the following sentences: (i) “In particular, the literature [xx] indicates the likely influence of socio-demographic factors, individual skills and family context.” (ii) “In the literature, some studies [xx] specifically linked socio-demographic determinants and nutrition behaviours, while others [xx] linked them directly with health outcomes such as overweight and obesity conditions”.

Response: Thanks for the very useful observation. The two sentences you reported were introductive to a more specific paragraph where the references are included. We specified that.

2.3. Please review: “Health literacy is a construct considered an important individual social determinant of health by many international bodies and studies” for “Health literacy is an important construct, as it is considered by international bodies and studies as an individual social determinant of health”.

Response: Thanks for the very useful suggestion. We modified the sentence accordingly.

2.4. To make the objective clearer, inform what the independent and dependent variables were.

Response: Thanks for the very useful suggestion. We modified the paragraph in order to make the objective clearer.

  1. MATERIALS AND METHODS

Table 1 - what does FAS mean?

Response: Thanks for your observation. We spelt out Family Affluence Scale (FAS) in Table 1.

  1. RESULTS

It is difficult to interpret Tables 3 to 8 without having text. Therefore, I suggest that the authors add a note to each Table to say which groups were compared for each variable.

Response: Thanks for the very useful suggestion. We added a footnote on Tables 3 to 8 to explain the categorical and ordinal variables included and make them easier to interpret. Following Reviewer 1’s suggestion, we also added a column specifying what level the blocks indicate.

  1. DISCUSSION

5.1. I suggest deleting the following part of the Discussion: “Children and adolescents’ overweight and obesity rates (…) to shape and influence children and adolescents’ behaviours” (first paragraph); as it was already presented in the Introduction section. Once that's done, the first paragraph of the Discussion section might look like this: “This study used a regionally representative Italian sample of 13- and 15-year-old students to measure and identify associations of participants’ different characteristics with their food consumption and BMI. The results confirm the importance of adopting socio-ecological approaches with regards to food consumption and eating habit of students, as well as their weight status. All explanation levels used in this study proved their association with outcome measures, although to different degrees".

Response: Thanks for the suggestion. We modified the paragraph accordingly.

5.2. Nardone and colleagues’ [28] findings do not support the result that students from families where at least one parent was not Italian were more likely to consume fruit daily. What can the authors say about this to include in the discussion (e.g., a speculation)?

Response: Thanks for the very useful observation. We better specified the differences between the two studies and added interpretations of such results.

5.3. Authors’ sentence: “This behaviour was associated with socio-demographic variables but not with health literacy and family variables". Add a comma (,) before the word “but”.

Response: Thanks for the very useful observation. We modified the sentence accordingly.

5.4. Authors’ sentence: “It seems that only social inequalities variables may influence daily breakfast”. Is there a why?

Response: Thanks for the very useful observation. We added some speculations on the reason.

  1. CONCLUSIONS

Authors’ sentence: “This study highlighted the importance of different factors associated with nutrition behaviours related to children and adolescents’ overweight and obesity”. Did the study involve people of childhood age?

Response: Thanks for the very useful observation. We removed the mention of children.

Round 2

Reviewer 2 Report

The authors did a good review following my suggestions. I only have a single additional review. The first paragraph of the Discussion is starting with references (i.e., [1–3][9,17–19]This study used...) and it has only a single sentence. Please review so that the set of references for it is allocated in its proper place and merge the first (i.e., This study used a representative regional sample...) and second (i.e., The results of this study confirm the importance of adopting a...) paragraphs. Thank you.

Author Response

We thank you for your suggestions, which have helped us improved the paper. The manuscript has been rechecked and the necessary changes have been made. The responses to all comments have been prepared and given below.

Comments:

The authors did a good review following my suggestions. I only have a single additional review. The first paragraph of the Discussion is starting with references (i.e., [1–3][9,17–19]This study used...) and it has only a single sentence. Please review so that the set of references for it is allocated in its proper place and merge the first (i.e., This study used a representative regional sample...) and second (i.e., The results of this study confirm the importance of adopting a...) paragraphs. Thank you.

Response:

Thanks for your positive feedback and your suggestion. Your comment is related to a mistake during the revision; we are sorry for that. The references have been deleted according to your previous comments, and the sentences have been merged.